



# Reusing Fe water treatment residual as a soil amendment to improve physical function and flood resilience

Heather C. Kerr[1], Karen L. Johnson[1], David G. Toll[1],

[1]Department of Engineering, Durham University, Durham, DH13LE, United Kingdom

*Correspondence to*: Heather C. Kerr (heather.kerr@durham.ac.uk)

**Abstract.** Soil degradation is a global challenge that is intrinsically linked to climate change and food security. Soil degradation has many causes, but all degraded soils suffer from poor soil structure. The UN's Sustainable Development Goals 12, 13 and 15 strive towards responsible consumption and production, building a zero-waste circular economy, achieving net zero by 2030 and reversing land degradation to protect one of our most valuable assets, soil. Global efforts to

stop and even reverse soil degradation require sources of both organic and inorganic materials to rebuild soil structure. The increasing global production of water treatment residual (WTR), an organo-mineral waste product from clean water treatment, means that the sustainable reuse of this waste provides a potential timely opportunity. Recycling or reuse of WTR to land is commonplace across the world but is subject to limitations based on the chemical properties of the material. Very little work has focused on the physical impacts of Fe-WTR application and its potential to rebuild soil structure

particularly improving its ability to hold water and resist the effects of flooding. This paper presents novel research in which the use of Fe-WTR and Fe-WTR/compost [1:1] co-amendment has shown to be beneficial for a soil's water retention, permeability, volume change, and strength properties. Application rates of WTR were 10 and 30% by dry mass. Compared to the control soil, co-amended samples have 5.7 times the hydraulic conductivity (570% improvement), 54% higher shear strength and 25% greater saturated water content. Single WTR amendment had 26 times the saturated hydraulic conductivity

(2600% improvement), 129% higher shear strength and 13.7% greater saturated water content. Data indicates that WTR can be added as a single amendment to significantly improve soil physical characteristics where shear strength and hydraulic conductivity are the most important factors in application. Although the co-application of Fe-WTR with compost provides a lesser improvement in shear strength and hydraulic conductivity compared to single WTR amendment, the co-amendment has the best water retention properties and provides supplementary organic content, which is beneficial for environmental

applications where the soil health (i.e. ability to sustain ecosystem functions and support plants) is critical. We develop the term 'flood holding capacity' to holistically describe the physical ecosystem services that soil delivers, which incorporates not only the gravimetric water content but the extra water storage potential due to increases in volume that occur in organic rich soils, the transmissivity of the soil (hydraulic conductivity) and the shear strength of a soil, which determines how well a soil will resist the erosive forces of water movement.

## 1. Introduction

Soil health can be defined as the ability of soils to deliver critical ecosystem services such as plant growth, water storage and carbon storage. As opposed to soil quality, soil health acknowledges that soil is a living system (Lal, 2016) and requires influxes of both organic and inorganic materials. A soil's water holding capacity, permeability, volume change and shear strength are fundamentally important for the delivery of these ecosystem services, as they control the flux of air and water

and turnover of soil organic carbon which feed the soil microbiome. Soil organic carbon is at the heart of good soil structure



which underpins soil health and can help to address climate change (Sommer and Bossio, 2014; Bonfante et al., 2020). The enhancement of soil water retention has long been addressed using organic amendments such as manure, biochar or compost owing to their beneficial impact on soil structure (Rahman et al., 2017; Zhou et al., 2019 and 2020). As such, the Paris Climate Change Agreement of 2015 launched the 4 per mille initiative where participating nations will seek to increase soil

organic carbon by 0.4% every year. Much research has considered the best organic materials to add to soil and it is generally agreed that 'active carbon' that can feed the soil microbiome allowing it to grow, is important to build soil organic carbon which helps store carbon. However, the water holding capacity of a soil is dependent not just on organic matter but on particle size distribution, which is largely governed by the inorganic mineral component of the soil. In addition, there is growing evidence that it is inorganic minerals which present the best opportunity to stabilise soil structure (Peng and

Smucker, 2007) stabilise carbon and help us achieve net zero (Tipping and Rowe 2019).

In this paper we do not consider the biological properties of the soil but present data on the physical properties (water retention properties, hydraulic conductivity and shear strength) of a sandy loam soil amended with both compost and an organic/inorganic 'waste' material, which is widely available around the world. The reuse of waste materials with large proportions of organic components is growing owing to building pressure to follow initiatives to both reuse and recycle

waste and improve soils (Hazbavi and Sadeghi, 2016; Krause et al., 2016). The worldwide supply of clean drinking water is an ever-growing demand, with up to 30% of the world still yet to have access to potable, treated water (Cotruvo, 2017; Turner et al., 2019). Treatment of source water is needed to remove harmful contaminants, pathogens, organisms and suspended solids, and usually involves the addition of iron (Fe) or aluminium (Al) salts to coagulate and flocculate these undesired particles for removal (Keely et al., 2014). The waste product of this initial process in the treatment of drinking

water is commonly referred to as water treatment residual, sludge or cake (henceforth referred to as WTR). Currently WTR is classified as a non-hazardous waste 'sludges from water clarification' (European Waste Code 190902, within 11. Common Sludges). From the point of view of rebuilding degraded soils, this waste provides a potential source of inorganic and organic matter since WTR contains both.

Many recent review publications such as Ippolito et al., (2011), Dassanyake et al., (2015), Mokonyama et al., (2017),

Odimegwu et al., (2018) and Zhao et al., (2018), and have discussed the wide range of global WTR properties, current disposal and reuse/recycling options for WTR and thus are not detailed here. Turner et al., (2019) provide the most recent and extensive review of potential reuse of WTR, however much of the focus is on Al-WTR. The potential for environmental remediation or WTR's use as a soil conditioner to improve soil function has been highlighted because of the high content of organic matter (8-30%) in addition to Al or Fe salts used (Makris and Harris, 2005). It is common practice for water

companies to spread WTR to land as a disposal method, with examples including Northumbrian Water [England] and Sydney Water [NSW, Australia] that dispose of 100% of their WTR via land spreading (pers comms, NWL 2019; Turner et al., 2019). Currently the land application of WTR is limited by the chemical attributes of WTR as it immobilises phosphorous (P) thus potentially reducing plant growth, in addition to potential for Al toxicity. Thus, WTR must be used alongside a potential source of P (Elliot and Demsey, 1991) or be spread at low rates of amendment by dry %wt. Despite

some promising early research e.g. Rengasamy et al., (1980), there is little continued focus on the effect of WTR addition on physical soil characteristics owing to the focus on the chemical implications of WTR spreading, and nothing on Fe-WTR owing to the prevalence of Al salts as a coagulant aid. To date there are no known publications that have holistically investigated the potential of Fe-WTR to improve not only physical soil characteristics such as water retention (relationship between suction and water content), but also permeability (hydraulic conductivity), volume change (soil shrinkage

behaviour) and shear strength. By understanding these three elements, one can assess the change in a soil's 'flood holding



capacity' due to amendment. Singular measures of water content such as field capacity (water content held against gravity, 2-3 days after a flood event) or saturated water content (mass of water/mass of solids when all voids filled are with water) are not sufficient to provide real indication of how a soil would respond to a flood event or prolonged periods of wet weather as other structural properties will also have significant influence. 'Flood holding capacity' is a term used to evaluate the

performance of amended soils, based on the water retention properties (gravimetric and volumetric water content of the specimens and hydraulic conductivity) in addition to the volume change characteristics of soil under climatic cycles and shear strength properties (Kerr et al., 2016). The effects of increasing soil organic carbon such as hydro structural stability, water retention and porosity are well known (Boivin et al., 2009), however without a holistic view of soil properties, the best amendment to address soil health in respect of increased flood events cannot be made. Owing to the nature of the

amendments used in this study, using gravimetric water content alone is problematic as inorganic and organic materials added to soil have high initial water contents (like WTR and compost) and low specific gravities, thus amended soil will naturally have an initially higher maximum gravimetric content (as shown by Moodley et al., 2004). Similarly, the problem with volumetric water content is that it is calculated using the ratio of volume of water to wet soil volume, or by using the gravimetric water content and dry density. As individual measurements, neither take convey any information on the increase

in volume of the specimen and therefore potential capacity to hold more water, particularly under flooded conditions (Boivin et al., 2009; Sollins and Greg, 2017). The existing system of measuring 'water holding capacity' fails to take into account the ability of a soil to change its volume and thus the effectiveness of amendments to soil will be discussed in reference to their contribution to flood holding capacity.

## 1.1 Water treatment residual characteristics

The physiochemical properties of many WTRs have been well characterised in different regions by a number of researchers (Elliot and Dempsey, 1991; Basim, 1999; Dayton and Basta, 2001; Dayton et al., 2003; Titshall and Hughes, 2005; Ippolito et al., 2011; Moreira et al., 2011; O'Kelly, 2016a;). It is generally considered to be an organo-mineral and the water content, mechanical and chemical properties of the sludge are entirely dependent on the procedures employed at each water treatment works (WTWs) and the nature of the local water source. The choice and dosage of coagulants and flocculants is based on

local water characteristics (turbidity, hardness, pH) and water treatment works use Fe, Al salts, polyelectolite (PolyDADMAC) or in some cases a combination of the two. Unprocessed WTR contains a fine solid fraction of between 2 and 4% (Dassanayake et al., 2015). This solid fraction comprises predominantly metal salt coagulant, organic constituents and other particulates from the catchment. Owing to the high volume of waste produced, water treatment works use various methods to dewater the sludge retrieved from settling beds to between 17 and 35% solids in order to make its transport and

disposal more economically viable, whereby the extent of dewatering is governed by the cost benefit (Li et al., 2016).
[Table 1]
Finlay (2015) provides a summary of characteristics of 9 WTRs from the NE UK (Table 1), which suggests that the nutritional elements of WTR are similar to soil but generally lower than other typical amendments such as biosolids and composts. Importantly the levels of N and P vary throughout the seasons which may be important when considering rate of

land application. UK figures are representative of WTR chemical attribute ranges from 62 other regions across the globe (Shen et al., 2019) such as the USA (Elliot and Dempsey, 1991; Makris et al., 2005; Agyin-Birikorang et al., 2007; Ippolito et al., 2009; Nagar et al., 2009;), Ireland (Babatunde et al., 2009), Czech Republic (Kyncl, 2008), Egypt (Mahdy et al., 2009), Australia (Oliver et al., 2011), South Africa (Titshall and Hughes, 2005), Belgium (Chiang et al., 2012) and Japan (Keeley et al., 2014).






### 1.2 The problem of WTR quantification

If we are to effectively manage the WTR waste stream and find suitable alternative reuse routes, the accurate quantification
of the waste stream is critical, especially with the growing demand for potable water (Ahmad et al., 2016) however, there
is little explicit information on WTR. The quantification of global production of WTR is an ongoing issue as highlighted
by Turner et al., (2019), and regularly cited publications on WTR production (e.g. Babatunde and Zhao 2007) are not
entirely accurate due to a lack of information on the water content of waste material, despite large differences in dewatering.
The difficulty in determining regional or temporal trends in WTR production lies in the accuracy of waste statistics, where
many national waste reports give total tonnes of waste for all water treatment which may often include sewage waste. For
example, The UK Statistics on Waste 2020 [2] data gives a figure for all water collection, treatment and supply, sewerage,
remediation activities and other waste management services (EU waste code 11, Common sludges), and specific data on
WTR can only be retrieved directly from water treatment works. An estimation by Dharmappa et al., (1997) provides a
prime example of this issue. They first estimated global WTR generation to be 10,000 tonnes/per day, a figure with no
annotation of water content. If taken as 10,000 tonnes of dried solids, the actual quantity of waste could be between 12000
tonnes and 500,000 tonnes owing to the potential differences in water content. If the 10,000 tonnes is a raw waste value,
the quantity of dry material in this waste sludge could be between 200 and 2000 tonnes due to differences in dewatering.
The WTR production values reported in more recent publications such as Zhao et al., (2018) and Turner et al., (2019) are
based on Babatunde and Zhao's (2007) extensive international review, however investigation into the source material of
this 2007 publication suggests that the annual figures of WTR production are in some instances incorrectly presented. Zhao
et al., (2018) presented a figure captioned 'Annual WTR generation (dry mass)', which expressed waste production in
tonnes of dry solids (henceforth TDS), by country and year. However, only the UK (2013) and Japan (2011) production
values in the figure were explicitly WTR reported by dry mass (TDS) according to the figure's source material. The Czech
(2016) WTR production presented by Babatunde and Zhao (2007) is correctly classed as sludge from water purification
(EU waste code 190902), however the water content is not stated by the Czech Statistical Office (2016) and thus assumed
to be TDS. The remaining source data is not explicitly WTR but all 'sludges from drinking water supply' (EU waste code
11.20), few of which have the water content noted. As standard practise, TDS should be supplemented with an annotation
of water content, or the total mass of wet waste must be supplemented with % of dissolved/dried solids or misinterpretation
is likely to continue. Inconsistencies in reporting thereby make the annual production trends difficult to quantify as the
water content is particularly important for this waste owing to highly variable solids content. This can be exemplified by
discrepancies in reported WTR production in the Netherlands; Babatunde and Zhao (2007) report Netherlands's 2000
production of WTR to be 34,000 TDS (unknown source), Zhao et al., (2018) report Netherland's 2014 production of WTR
to be 382,000 TDS based on the DWMA (2017) report, which does not contain information on the water content and
Mokonyama et al., (2017) state that Netherlands produced 170,000 TDS in 2017, where their description refers to the sludge
that is formed during the production of potable water. There is clearly a misunderstanding in the water content or type of
waste, as an increase by an order of magnitude in 10 years is unlikely. AquaMinerals (2019) work with all water treatment
works in the Netherlands to find suitable recycling streams for water treatment waste and maintain detailed annual records
of waste production. In 2019, 17,984 TDS of WTR were produced (pers comms with Director of Aquaminerals) and
therefore the production estimate of 34,000 TDS in 2000 by Babatunde and Zhao (2007) may realistically be TDS as
reported, and the reduction of WTR production between 2000 and 2019 reflects ongoing refinement of water treatment in



the Netherlands. However, the 382,000 TDS reported to be produced in 2014 by Zhao et al., (2018) and 170,000 TDS

produced in 2017 by Mokonyama et al., 2017 are therefore very likely to be 'wet' tonnes or include other or all other wastes

from every stage of potable water treatment. Therefore, it is with extreme caution that production estimates can be analysed,

as although reports may accurately report total waste or specifically WTR by dry mass/TDS, the actual volume of waste

that must be disposed of will be 5-20 times the dry mass reported, given sludges have between 5% and 30% total dissolved

solids. In order to build a sustainable, circular economy that promotes the recycling of waste materials in line with SGDs,

the explicit total volume of waste and the volume of dried solids produced are critical pieces of information. Without

accurate statistics and reporting of wastes that may include large volumes of water, one cannot grasp the scale of increasing

production of waste materials and thus the land mass required to recycle WTR.

### 1.3 Land application of WTR

There are a considerable number of potential applications for the reuse or recycling of WTR that have been extensively

covered in other publications and thus are not discussed here (Ippolito et al., 2011; Zhao et al., 2018; Turner et al., 2019).

The existing research on land application of WTRs has focussed on the chemical implications of its application (Moodley

et al., 2004) with many investigating the effect of Al-WTR amendments on crop growth owing to potential toxicity of Al,

the presence of other potentially toxic elements (PTEs) and phosphorous immobilisation. Despite immobilising P, WTR

reduces mobile PTEs and can be safely be applied to land.  Zhao et al., (2018) provide an extensive review of plant trials

using both Fe and Al WTR as a soil amendment and found no standard plant response to WTR addition. For example,

Oladeji et al., (2007) and Oladeji et al., (2008) reported alternate plant responses despite otherwise identical parameters at

an application rate of 10- 25 g/kg by oven-dry mass.

The need to investigate the physical effects of WTR amendment on soil characteristics was noted as early as 1980

(Rengasamy et al., 1980), however as discussed by Moodley et al., (2004) there is a scarcity in research on physical changes

resulting from amendment. To date only a few studies have explicitly explored the effect of Al-WTR on physical soil

characteristics, specifically their water retention (relationship between water content and suction) and permeability

(hydraulic conductivity), and none to the authors knowledge that have investigated Fe-WTR. Some publications have

mentioned changes in water retention, bulk density (dry mass/total volume), and permeability, but these are noted as

secondary findings and are not discussed at any length. Rengasamy et al., (1980) is the only published study investigating

the effect of WTR amendment on shear strength despite this being a critical parameter for the resilience and stability of

soils under saturation. The specific material shear strength characteristics of WTR have been investigated extensively but

only for their use in construction materials (Hegazy et al., 2012; Rodrigues and Holanda, 2015: Gomes et al., 2019; Liu et

al., 2020).

### 1.4 Use of WTR to improve physical soil characteristics

Using WTR to improve fundamentally important soil characteristics (water retention, conductivity and shear strength) has

the potential to address two goals; firstly, the need for resilient and regenerated soils required under many environmental

directives (SDG, WFD, etc) and secondly the many directives to reduce, reuse, recycle. Aquaminerals (Netherlands) have

shown that there is an opportunity to recycle or reuse 100% of WTR produced each year, and further research to support

the reuse/recycling of WTR to land is of particular importance to encourage higher rates of reuse rather than disposal to



landfill. This is particularly important under enhanced climate change with growing extreme events (Madsen et al., 2014) and is reflected by the targets of SDGs to look after soil ecosystem services.

As a prelude to subsequent literature discussion on the effect of WTR application, amendment rates are typically presented as Mg/ha or as a wt/wt% of dry soil mass in literature. For ease of comparison, the application of WTR per hectare been assumed to be incorporated to a depth of 0.2 m into a control soil of 1.2 g/cm$^3$ dry density, giving an amended soil volume

of 2000m$^3$/ha, or 2400Mg of soil/ha. As discussed earlier, many publications make no note of water content of WTR, thus all calculations assume WTR is in a raw dewatered format (20% solids) unless stated otherwise, and amendment is presented in tonnes of dry solid (TDS) per tonne of dry soil as a percentage (%). More than 40 years ago, El-Swaify and Emerson (1975) found that Al and Fe hydroxides in WTR act as cementing agents between soil particles, which imparts fundamental changes to important soil physical parameters, by reducing swelling and increasing aggregate stability (Elliot and Dempsey,

1991). Rengasamy et al., (1980) reported 18-85% increase in water holding capacity with extremely low rates of Al-WTR application [20Mg/ha = 0.002% wt/wt] whilst also reporting increased aggregation. Elliot and Demsey (1991) identified potential for sludges to significantly to alter soil aggregation, permeability and modify water retention properties based on the high proportion organic matter in the material. More recently, studies have also noted these improvements using Al-WTRs, but these physical parameters have not been the focus of the study (Basta et al., 2000; Dayton and Basta, 2001).

Moodley et al., (2004) specifically investigated the water retention and hydraulic conductivity of soil amended with up to 10.6% wt/wt amendment [1280Mg/ha] air-dried WTR and found increased saturated water content at highest application rate and a shift in the water retention curves was observed due to addition of fine particles, but the readily available water (-10kPa to -100kPa) was unchanged. Three years after the start of the trial, all soil water retention curves (SWRCs) were almost identical to the control which may indicate that these beneficial changes are short term. Moodley et al., (2004)

observed a 9-10 fold increase in hydraulic conductivity with WTR addition due to the creation of preferential pathways (large pores and surface channels). WTR's improvement of hydrological conductivity was attributed to high stability and limited swell of WTR aggregates, which reduces pore blockages shown in control soils. The addition of highly porous WTR reduced the bulk density of the soil, an effect strongly correlated with the proportion of amendment. Moodley et al., (2004) summarise that despite increased water retention and conductivity at the highest rate of application, there is no additional

benefit for crops as plant available water remains unchanged. To date Rengasamy et al., (1980) is the only published study that has directly investigated WTR's effect on shear strength, reporting a decrease in response to addition of WTR (Herselman, 2013). To the authors' knowledge there has been no published research into the long-term effects of Fe-WTR application. Ippolito et al., (2009) investigated the long-term effect (15 years) of Al-WTR and biosolid co-amendment at up to 21Mg/ha (no incorporation into the soil profile) and found minimal disruption of soil chemistry, microbial diversity

or plant nutrient levels. Ippolito et al., (2009) also suggest that the single application of Al-WTR would pose little threat to plant or soil biology and provide long term solutions for soils with excess P. Mukherjee et al., (2014) provide another example of 'long-term' application of Al-WTR to amend soil (0.5% wt/wt) and found little change in soil physiochemical proprieties and greenhouse gas emissions compared to the control soil over two years, although they did report increases in hydraulic conductivity (7.1- 7.5 uSm/s) and penetration resistance increase of 87%. They suggest the need for additional

research that uses higher application rate and a longer-term research period before providing further conclusions.

Research has shown that WTR amendment has the potential to increase water retention under drier conditions and increase water held at saturation. In addition, the infiltration and saturated hydraulic conductivity have been reported to increase significantly. These findings are important as these parameters alone indicate that WTR has the potential to increase a soil's ability to take up and store flood water. However, with the exception of a study by Kerr et al., (2016), no research has



holistically investigated the effect of Fe WTR soil amendment on water retention and parameters that are important for flood resilience such as permeability and shear strength, despite these important physical changes having been noted in previous research. Elliot & Demsey (1991) identified the potential for WTR to alter each of these parameters but did not consider their combined effect.

In a recent publication, Kerr et al., (2016) tested the ability of air-dried Fe-WTR and an air-dried Fe-WTR and compost co-
amendment to change the physical properties of a sandy loam soil. WTR was applied at between 30 and 50% (wt/wt dry mass), which are in the extreme upper limits of realistic field application to produce small remoulded soil specimens. The co-amendment of Fe-WTR with compost was the focus of the study, however single WTR and single compost amendments were included as additional controls, as although the effects of compost on water retention properties are well understood, the effect of WTR amendment is not. Many studies have not considered WTR as a single amendment owing to its limitation
on plant growth, however the physical effects of WTR amendment on soil structure is an important line of enquiry. Amended soil samples were tested for their water retention under flood conditions (maximum gravimetric water content), and shear strength (inferred by fall cone penetrometer). The most important finding of this study was that at saturation, the 30% co-amendment held almost as much water [0.4 g/g] as the single compost amendment at the same rate of application [0.43 g/g]. Compared to the maximum achieved by soil alone [0.32 g/g] this is an important improvement. This provides
significant rationale for exploring co-amendment of WTR and compost as an alternative to singular organic amendments typically used to improve soil structure. The single amendment using WTR did not improve the maximum gravimetric water content of the soil compared to the control, although it did improve the rate of water ingress by up to 42% in the first 24 hours compared to the control soil. Fall cone penetrometer (BS 1377–2:1990) testing indicated that the addition of WTR had considerable potential to increase the undrained shear strength of specimens but owing to large error in results, the
effect was inconclusive.

**2. The use of Fe-WTR and compost to improve flood holding capacity**

To investigate the physical effects of Fe-WTR amendment on a sandy loam, remoulded amended soil specimens were prepared for laboratory testing. Testing parameters included water retention (gravimetric and volumetric water content at saturation), saturated hydraulic conductivity, shear strength and soil shrinkage behaviour (volume change). Four
amendment types were tested: [1] control soil, [2] Fe- WTR (as raw and air-dried) [3] compost and [4] co-amendment using WTR and compost. Each amendment type was tested at application rates of 10 and 30% wt/wt by dry mass.

**2.1 Materials**

Well characterised topsoil was retrieved from a local farm (Nafferton Farm, NZ065656), Fe-WTR obtained from Mosswood Water Treatment works (Northumbrian Water Ltd) was added as an amendment to soil as both as the raw, dewatered
material retrieved from the water treatment plant, subsequently named WTRw [~20% total dissolved solids], and an air-dried version of the same material [0.2 g/g gravimetric water content), named henceforth as WTRd. As WTRs exhibit a wide range of physiochemical properties determined by local factors, and therefore extensive characterisation of the Mosswood WTR (Kerr, 2019) was carried out prior to soil testing. Commercially available compost was used as received from the supplier (PAS100 compost, 100% recycled green compost product derived from surplus garden trimmings and
local authority recycling contracts). Remoulded soil specimens were compacted dynamically to a density of 1.75g/cm³ at a water content of 0.175 g/g (based on optimum water content of the control soil). As per Kerr et al., (2016), single





amendments of compost and Fe-WTR were tested in addition to the co-amendment, in order to investigate the individual effect of the amendment on the control soil. The application rate of WTR was chosen based on the maximum spreading thresholds of current UK legislation of 3000 tonnes/ha/year, which equates to ~50% amendment (%wt/wt_by dry mass)

with the assumptions made in section 1.4 that incorporation depth is 0.2m and dry density of the soil is 1.2 g/cm³. Realistic typical deployment of 100tonnes/ha equates to just 0.04% amendment %wt/wt by dry mass (NWL 2019, pers comms).

**2.2 Material properties**

Table 2 summarises the material properties of soil, WTRd, WTRw and compost, characterised using British Standards BS1377. Particle size distribution testing was not achievable for WTRw due to issues with settling fine particles in the

hydrometer. The small pycnometer method was used to characterise the particle density of WTRd and compost. Following Weindorf and Wittie (2003) and O'Kelly (2016b), hexane replaced water in the determination of compost particle density to facilitate submersion of very low-density material. Both compost and WTRd were de-aired within the flask using a vacuum pump for a period of 3 days before particle density. This method was adopted following Basim (1999) in their extensive characterisation of WTR and organic soils, as heating the contents for air removal would result in the loss of

volatile solids and organic fractions.

[Table 2]

**2.3 Sample preparation and testing**

Preliminary data (not presented) indicated that specimens produced using dynamic compactive effort had considerable differences in density, depending on their amendment proportions and water content, and as such made comparison of water

retention characteristics between different amendments very difficult to assess. As such, the dry density [mass of solids (g)/volume of specimen (cm³)] and gravimetric water content [mass of water/mass of dry solids] were used as controlling factors in specimen preparation in order to provide more comparable data between different amendments. By comparing the performance of amendments based dry mass per unit area, the density as a controlling factor on water retention is removed and an assessment can be made on the retention properties of the material. The drawback of this method is that

compost must undergo greater compressive efforts to achieve the required dry density owing to compost's low particle density (1.675 g/cm³) and thereby reduces the macro porosity through which it achieves good water holding capabilities, although Peng et al., (2012) found that soil volume change characteristics were independent of soil compaction intensity. This isn't as problematic for soil and WTR as their particle densities are more similar (2.65 g/cm³ and 2.11 g/cm³ respectively). The compost within specimens was able to swell and regain their structure once wetting was initiated, thus

reducing the initial effect of compaction during preparation.

Cylindrical specimens 38 x 76 mm were compacted using a static press to a dry density of 1.75 g/cm³ at a water content of 0.175 g/g. WTRd was prepared by air-drying and sieved to 6.3 mm before addition to soil. WTRw was added to soil as a dewatered sludge (20% solids) and mixed with soil before the amended material was dried to 0.175g/g. As such, both WTRd

and WTRw are air-dried forms of WTR, but amendment using WTRw produces an amendment mixture with greater distribution of fines owing to the preparation procedure. Once extruded, the sides of the specimens were coated with liquid latex and open ends of the specimen were wrapped in fine material to reduce the loss of fines. Amendment proportions were as follows and were mixed based on the dry mass of each component material. 100% soil (control), 30% single amendment





of WTRw, WTRd, compost, or 30% co-amendment (1:1), 10% single amendment of WTRw, WTRd 10% compost or 10%
co-amendment (1:1). Twelve replicates were produced for water retention testing, and three replicates for trixaxial testing
(shear strength and hydraulic conductivity). Specimens were subjected to climatic cycles as per Kerr et al., (2016). Water
ingressed through the base of the sample, and once mass negligibly increased the specimens were flooded for a period of
72 hours [Fig. 1]. The duration of wetting was 7 - 14 days depending on the amendment type. After the 72-hour period of
flooding, specimens were dried for two weeks before the second wetting was initiated. As such the water content of

specimens was variable after drying. The mass and specimen volume were measured using a standard mass balance ±0.01g
and digital callipers, through two wetting, flooding and drying cycles. Specimens were tested in triaxial apparatus 4 weeks
after preparation, but had not undergone any climatic sequences before testing. The saturated hydraulic conductivity of
specimens was tested during consolidated undrained triaxial testing (BS 1377), once the sample had been consolidated.
Three specimens of each amendment were subsequently sheared each at the specified cell pressure (25, 50 and 100 kPa).


[Figure 1].

### 3 Results and Discussion

As soils that have been compacted or remoulded have been found to reach equilibrium after three to five wetting/drying
cycles (Tripathy et al., 2002), analysis of data was conducted on the second wetting and drying cycle to investigate the

initial effects of WTR amendment. Further climatic cycles were not achieved due to deterioration of specimens. Numerical
testing for statistical significance was conducted using Mann Whitney test assuming non-parametric data. In general, there
were no statistical differences between amendment using WTRd or WTRw for much of the data presented below, despite
numerical differences in the averaged values of these amendments. This may reflect heterogeneity between specimens
amended with WTRd or WTRw rather than a fundamental difference in material properties as a result of the addition of

WTR in a raw or air-dried format.

### 3.1 Water content relationships

All amendments yielded statistically significant ($p<0.05$) higher gravimetric and volumetric water content at saturation
compared to the control soil (with the exception of 10% WTRd co-amendment). Fig. 2 shows that specimens with the
highest ratio of amendment had the highest water retention at saturation. Compared to the control, the 10% amended soil

had higher gravimetric water contents of between 5.4 and 17.8% (higher volumetric water content between 11.3 and 34.4%).
30% amended soil showed increases of between 11.2 and 32.4% GWC (12.8 and 43.6% VWC) compared to the control.
This indicates that the degree of change is proportional to the ratio of amendment. An understanding of the soil water
retention curve is needed quantify suction values at both low and high degree of saturation, in order to determine how much
of this excess water is plant accessible.

[Figure 2]

The key findings for water retention are that: all single amendments of WTR significantly increased the gravimetric and
volumetric water content at saturation compared to the control soil. The differences in GWC/VWC were; 10WTRd [increase
of 11.7/21.9%], 10WTRw [increase of 9.5/11.8%], 30WTRd [increase of 22.3/23.5%], 30WTRw [increase of 11.2/12.8%]).
Compost only amendments have higher GWC and VWC at saturation compared to single WTR or co-amended specimens

at the same ratio. This is an expected finding owing to compost's well known water retention properties (Kay 1998). The



use of co-amendment improves both the GWC/VWC compared to the control by 24.7GWC/43.6VWC% at 30% amendment, and 10.9GWC/11.3VWC% at 10% amendment.

These findings are important for the potential of amendment to increase flood holding capacity, as it is clear that both co-amendment and single amendments of WTR increases the water held at saturation compared to the control soil thus increasing the amount of water held in the soil before the initiation of surface run off. Although the addition of a single compost amendment produces the best improvement of water retention at saturation, these results show that the addition of WTR does not hinder the ability of organic amendments to retain water.

**3.2 Soil volume change**

Soils with high organic content have more pronounced hysteresis than inorganic soils which are stiffer in comparison (Peng and Horn, 2007), which suggests that amending soil with WTR would mean a greater volume change during climatic cycles in response to wetting or drying. However, Elliot and Dempsey suggest that WTRs reduce swelling and increasing aggregate stability (Elliot and Dempsey, 1991) owing to their chemical attributes. The volume change properties of a material are viewed differently by various disciplines; civil applications require soils with low organic fractions and little volume

change. In this environmental application, the increase in volume and reduction in density of soil is important to incorporate as much water as possible into the material when wetting occurs. However, a successful amendment to soil in this respect is one that encourages swelling to incorporate while limiting shrinkage during drying phases. Changes in soil volume during wetting and drying can occur due to external stresses (i.e. compaction or load) but owing to the free swelling and shrinkage of specimens, the volume change is related to the internal stress and suction (Peng and Smucker, 2007).

[Figure 3]

In Fig. 3, the relationship between dry density and water content at saturation for amended soils indicate that soils with the greatest organic content undergo the greatest swelling during wetting. Specimens were initially prepared at $1.75 \text{g/cm}^3$, and thus the change in dry density is an indicator of the ability of the material to swell upon wetting. Figure 3 also shows that the water retention at saturation appears to be loosely correlated to the dry density of the specimen at saturation, as with

lower density indicates a greater proportion of pore space. The control soil had the highest dry density [$1.41 \text{g/cm}^3$], lowest GWC [0.314] and VWC [0.444]. In contrast the 30% compost amendment had the lowest dry density [$1.17 \text{g/cm3}$] and highest water content [0.417GWC/0.492VWC]. The correlation between dry density and maximum gravimetric water content is much stronger than the trend shown for volumetric water content. As discussed in section 1, volumetric water content as a measure doesn't convey specimen volume change. For example, 10C shows a low volumetric water content

compared to other 10% amendments, but this is because the soil has increased in volume and decreased in dry density, and therefore the ratio of volume of soil to volume of water has reduced.

Further investigation is needed to characterise volume change characteristics, particularly testing over more climatic cycles to reach equilibrium is required to evaluate the true effect of amendments on the swelling and shrinkage properties of the soil. Amended samples undergo a large degree of expansion when wetted, but the shrinkage (data not presented) is greatly

reduced compared to the control soil, indicating that the structure amended specimens is more stable than the control soil. This is important for the application of amendments in areas where large volume change is undesirable, but less so for flood resilience. The relationship between the parameters of dry density, water content and volume change for specimens with the addition of highly compressible material (compost) needs greater exploration, as the preparation conditions, i.e. the use



of dry density as a control, clearly have a fundamental impact on the performance of the material in terms of the water
retention and volume change.

### 3.3 Hydraulic conductivity

Figure 4 shows the variation in saturated hydraulic conductivity ($k^{sat}$) at three different confining stress levels (25 kPa, 50 kPa and 100 kPa), using a pressure difference of 15 kPa to generate flow. In general, it is clear that at 25 kPa all amendments improve the $k^{sat}$ compared to soil alone, with the exception of 30% compost, as shown in Fig 7 [right]. Single amendments
of WTR increase the $k^{sat}$ to the greatest extent [5.7-26.3 fold increase]. Compost amendments provide little improvement in $k^{sat}$ at 25 kPa and at higher cell pressure the presence of organic matter causes a reduction in $k^{sat}$. The ability of the co-amendments to improve $k^{sat}$ is dependent on the amendment ratio, where higher amendment proportions show the greatest improvement. The $k^{sat}$ of specimens is lower at greater confining pressure, as increasing the confining stress causes a reduction in porosity, making water flow more difficult. This is particularly evident for the compost containing amendments.
The control soil has a $k^{sat}$ of $6.73 \times 10^{-7}$ m/s at 25kPa, which is classed as moderate but typical of sand, silt and clay mixtures. WTR amendments have very high $k^{sat}$ in comparison with the control soil at 25 kPa; 30%WTRd has the highest $k^{sat}$ [$1.77 \times 10^{-5}$ m/s], which is typical of clean sands, followed by 30% WTRw [$7.2 \times 10^{-6}$ m/s], 20% WTRw [$5.3 \times 10^{-6}$ m/s], 10WTRw [$4.1 \times 10^{-6}$ m/s], 30% coWTRw [$3.9 \times 10^{-6}$ m/s) 20% WTRd [$3.8 \times 10^{-6}$ m/s]. The value obtained for 10%WTRd was $6.5 \times 10^{-7}$ m/s indicating that this result may be an error. The $k^{sat}$ of WTR amended specimens is therefore 5.7 to 26.3 times greater
than the control soil (570-2600% improvement). WTRd amendment contributes irregularly shaped coarse particles with a broad range of particle size to the control soil, whereas WTRw amendment adds a lower proportion of small irregularly shaped particles and a large proportion of very fine material [<75um]. Both of these additions improve the soil structure, reduce the bulk density of the amended specimen and increase pore space as a result of irregular particle shape. As one WTR type does not consistently result in an improved conductivity over the other, this difference may also be due to
heterogeneity of specimens rather than fundamental difference in material properties. These results indicate that the addition of WTRw and WTRd has a significant impact on the $k^{sat}$ of the soil, and compost addition dampens this effect. The high $k^{sat}$ of amended soils indicates that a soil would be able to transmit water through the soil profile rapidly and avoid water pooling at the surface during high intensity events which is particularly important for reducing erosion at the soil surface.

[Figure 4]

### 3.4 Shear strength

Amended soil specimens were tested in a triaxial apparatus to determine any changes in strength due to the addition of WTR or a WTR/compost coamendment whilst in a saturated state, an important measurement to understand how well a soil will cope with shear stresses when flooded. It is clear from Fig. 5(a) that at low confining pressure (25kPa) the amendments
with WTR alone (30WTR) show the greatest improvement compared to the control soil. Soils amended with 30% compost (30C), and co-amendments with 15% compost and 15% WTR (30coWTR) show an improvement in strength compared to the unamended soil (100S) when tested at 25 kPa. It can be seen from the stress paths in Fig 5(b) that the paths for the amended soils shift to the left compared to the unamended soil (100S), indicating more pore water pressure development. This might be expected for the compost amendments, due to the more compressible nature of the compost. It is interesting
to see this is also true for the WTR alone. It seems that the amendment with the wet WTR (30WTRw) shows the greatest





improvement (Fig. 5(a)), but the sharp kink in the stress path in Fig. 5(b) might suggest that the test using dry WTR (30WTRd) developed a localised shear surface during testing, that might have restricted the ability to gain in strength. Therefore, the difference between 30WTRw and 30WTRd could be due to heterogeneity rather than a fundamental difference in behaviour.

[Figure 5]

At the higher stress level of 100kPa (Fig 5(c)), the amendment of compost alone (30C) shows no improvement compared to the unamended soil. However, the co-amendments with 15% compost and 15% WTR (30coWTR) do show significant improvements in strength. Again, the greatest improvements are for the WTR alone (30WTR). There is no major difference whether the WTR is wet or dry. Again, the stress paths in Fig. 5(d) show that all the amended soils develop much greater

pore water pressure than the unamended soil (100S), indicating a more compressible response. This increase in strength with the addition of WTR may be due to physical changes in soil structure owing to the irregular shape of WTR particles, or through chemical contributions of the flocculant and coagulant aids that make up WTR. Further investigation is needed to understand the strength mechanics of amended soil, particularly after specimens have been subjected to climatic cycles.

**4 Conclusions**

The amendment of a sandy loam soil with either Fe-WTR or with Fe-WTR and compost co-amendment has significant impacts on a soil's flood holding capacity, which incorporates the parameters of water retention, hydraulic conductivity, volume change and shear strength. This holistic view of how an amendment affects the properties of a soil is essential if the amendment is going to effectively used to improve critical services of the soil. The following generalisations can be made on the changes in physical soil parameters; the single addition of WTR results in an increase in water content and reduction

in dry density at saturation, increases the saturated hydraulic conductivity and shear strength of the soil compared to the control. The use of compost as a single amendment yielded the best improvements in water retention properties however the negative effects of this amendment on saturated hydraulic conductivity and shear strength means that it does not impart the properties required for a flood resistant soil. However, the addition of compost and WTR as a co-amendment combines the beneficial properties of both materials, resulting in higher water content at saturation compared to both the control soil

and WTR alone, whilst also increasing saturated conductivity, increasing the volume change (reduction in dry density) and the shear strength of soil (albeit to a lesser extent than WTR amendment). As such, the co-amendment offers an advantage over the use of single amendments of WTR or compost as each have their own drawbacks of application (concerns on PTEs and P immobilisation of WTR, and lack of shear strength and hydraulic conductivity of compost). Data presented suggests that recycling WTR can provide a route with which to address critical SDGs to rebuild and regenerate soils whilst

encouraging a circular economy. Finding an optimal amendment balance for amending soil means that with increasing clear water production, water treatment companies are able to fulfil their requirement to recycle waste while benefiting degraded soils. Further research is needed to establish the optimal amendment ratios to provide the most beneficial improvement to physical soil parameters for ecosystem functions, particularly the flood holding capacity of an amended soil. This includes monitoring the water retention and volume change of specimens over >4 climatic cycles, an assessment of unsaturated

hydraulic conductivity, and an assessment of shear strength of amended soils after several climatic cycles. An understanding of the soil water retention curve would aid in further discussion of these changes and allow us to fully understand WTR's effect.





*Data availability* Full data sets are available from FigShare. DOIs available upon request.

*Author Contribution* Research was conceptualized and conducted by H.Kerr under the supervision of K.Johnson as part of
an EPSRC funded PhD. D. Toll aided in investigation of triaxial testing in both concept and review of data and writing for publication.

*Acknowledgements* Data in this publication was drawn from research was funded by the EPSRC as part of a PhD project at

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



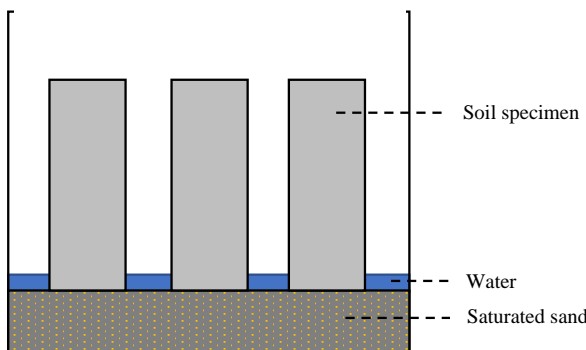

**Figure 1: Soil specimens with latex coating, situated on saturated silica during for the primary wetting sequence.**







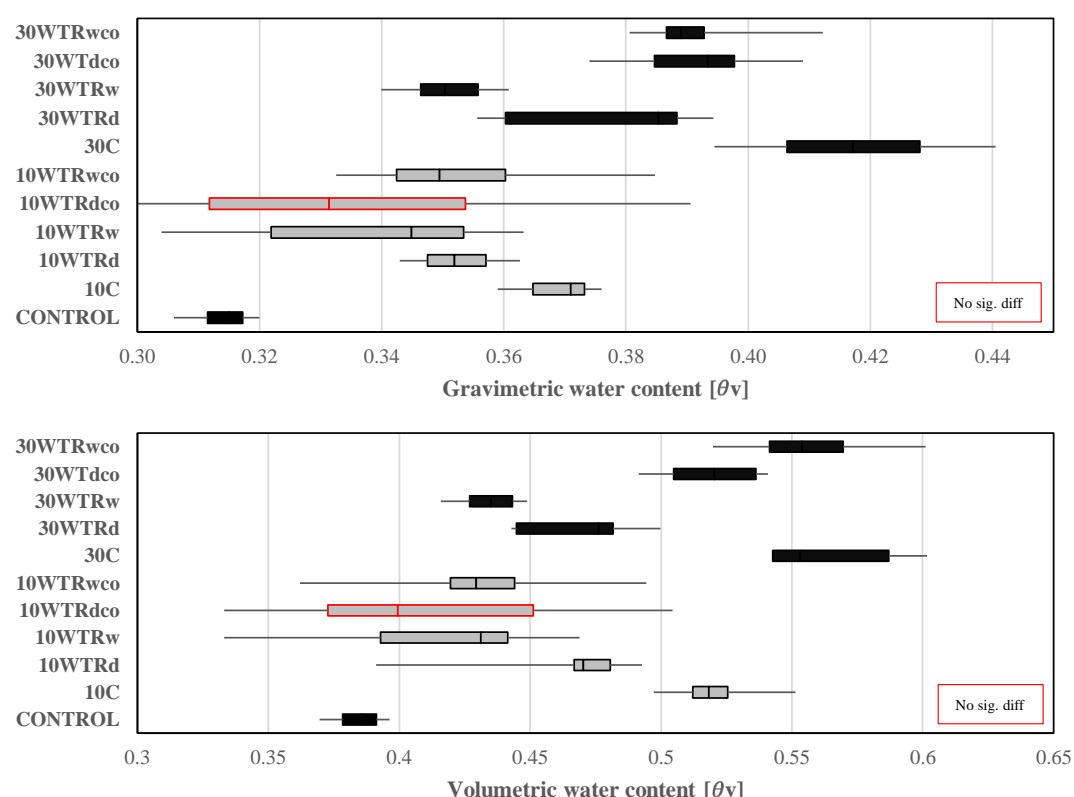

**Figure 2: Box plot of maximum gravimetric water content ($\theta_g$) and volumetric water content $\theta_v$) of specimens at saturation after second flooding event.**




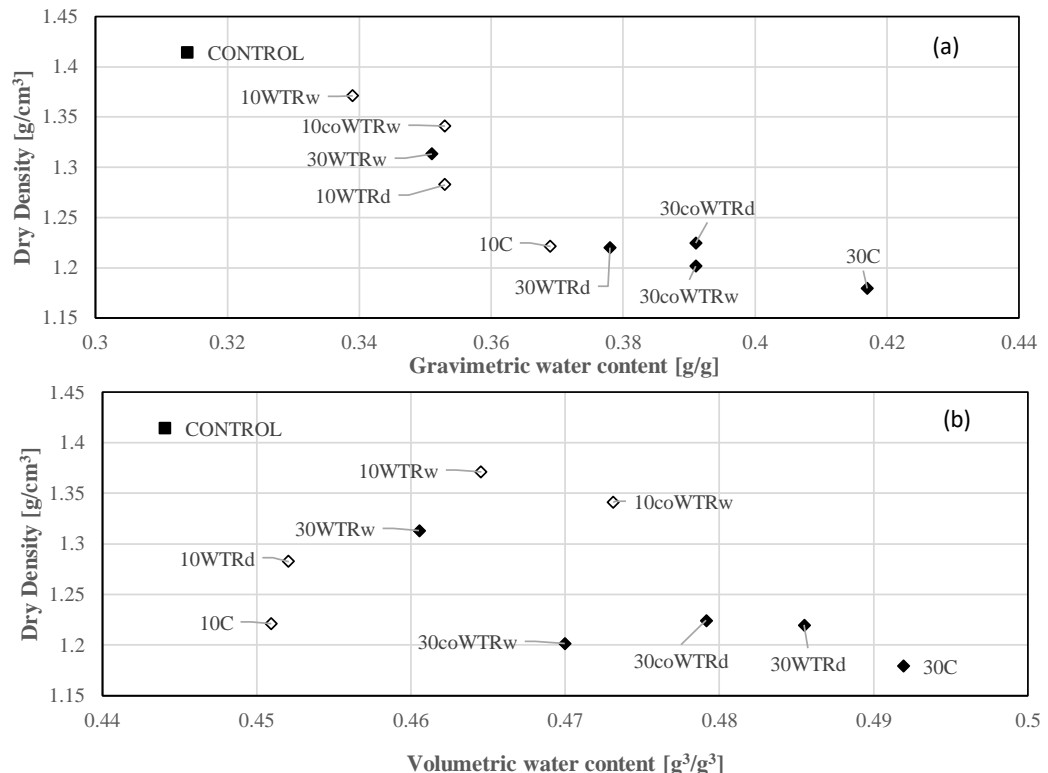

**Figure 3: (a) Relationship between dry density and saturated gravimetric water content, (b) relationship between dry density and saturated volumetric water content**







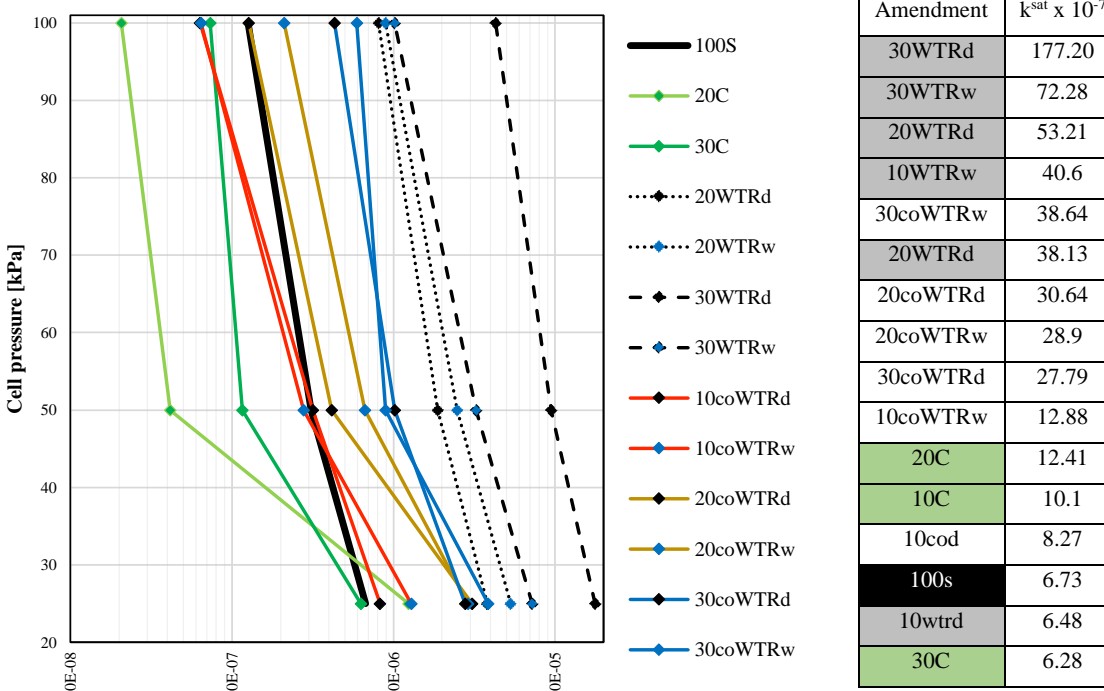

**Figure 4: [left] Saturated hydraulic conductivity of samples conducted in a triaxial cell at pressures of 25, 50 and 100 kPa, on a log scale. Samples 10C, 10WTRd and 10WTRw were only tested at 25 kPa and do not feature on the graph; their k values were 1x E-06, 6.48 x E-07 and 4.06 x E-06 m/s respectively. n = 1 for all samples. [right] ranked conductivity values for all specimens at 25 kPa.**






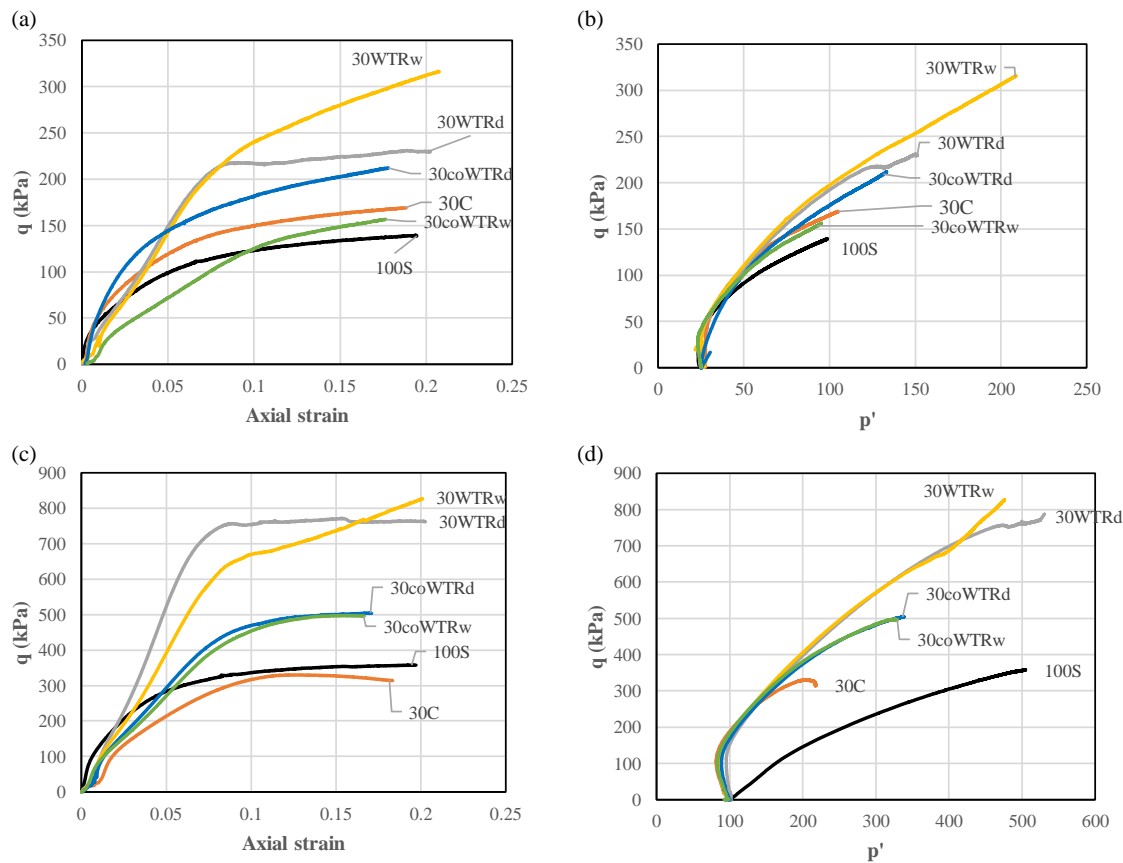

**Fig. 5(a) and (b) show the stress-strain response and stress paths for undrained triaxial tests carried out a confining stress of 25 kPa, and (c) and (d) shows the same information for tests carried out a confining stress of 100 kPa.**







| Parameter | Range | Soil/Biosolids typical | Parameter | Range | Soil/Biosolids typical |
|---|---|---|---|---|---|
| Dry solids | 2-28% | - | Mn* (mg/kg) | 370-5100 | 1300/200 |
| pH | 4.09-8.6 | - | Total N % | 0.51-1.1 | 0.5/4 |
| EC (us/cm) | 30-405 | - | C:N | 15.5-39 | 10 |
| Fe* | 0.8-41% | 4/1.5% | P* (mg/kg) | 4-1528 | 1000/25000 |
| Al* | 0.21-21% | 7.1/0.5% | K* (mg/kg) | 170-3900 | 640/3000 |
| LOI550 | 36-70% | 5/70% | Mg* (mg/kg) | 170-2900 | ~/2000 |
| Total C | 13-26% | 3/40% | | | |

**Table 1: Summary of WTR characteristics NE UK (Finlay 2015) *pseudo total metals measured by aqua regia digestion. Total and inorganic carbon measured by Thermo TOC1200. Due to <0.1% inorganic C, Total C represents organic C content of WTRs.**

| Property | Nafferton Farm soil | Fe- WTR | Compost |
|---|---|---|---|
| Particle size distribution Wet sieve analysis BS1377 | Sandy loam 61.8% sand, 25.1% silt, 13.1% clay | 1nm-1 μm fractions 95% passing 74um sieve (Basim, 1999). | n/a |
| Gravimetric water content [g/g] BS812 | 0.16 g/g | WTRw: 4.94 g/g (17% dry solids) WTRd: 0.18 g/g | 0.55g/g |
| Particle density *Small pycnometer method* BS1377 | 2.65 g/cm$^3$ [+] | 2.11 ±0.81g/cm$^3$ | 1.675 ±0.33g/cm$^3$ |
| Pre-treatment | Sieved to 6.3 m | WTRd: Air dried and sieved to 6.3 mm (dry) WTRw: No treatment | Large fragments removed. |
| Chemical properties | pH 7.5[+] LOI550 3.96%[+] TOC 2.3%[+] | pH 4.7 ±0.5* [ISO 10390,2005] EC 239+168* [ISO 10390,2005] Fe 31%* LOI550 48±2.7%* [BS1377] TOC 27.9%* [ECS 4010] | pH 8.1[+] LOI 13.9%[+] TOC 14%[+] |

**Table 2: Material properties and preparation. Data annotated* are sourced from Finlay (2015). ICP-MS Pseudo-total metal concentration for Fe was carried out by NWSS labs [24]. Data annotated [+] were obtained by DETS (2018).**
