# Peer review of "Reusing Fe water treatment residual as a soil amendment to improve physical function and flood resilience"

_SOIL, 2021_

## Author Response (AR1)

**Author's response**

Dear Editor and reviewers.

We have addressed the comments from each reviewer in turn and included the corrections in the manuscript as requested. The extensive introduction has been reduced by half to only include the relevant information and restructured so that a better narrative is told. We remove the idea of 'holistic' as this was confusing to the reader and have instead highlighted that we have only investigated the physical parameters as the chemical/biological ones are covered elsewhere. Nonetheless we briefly discuss (as requested) the potential effect of WTR addition on the chemical/biological factors and state the gaps in knowledge between Al and Fe WTR. We identify that physical effects of WTR amendment are very sparse, particularly with Fe WTR, thus providing rationale for the research.

List of corrections:

1. Abstract has edited to remove the least relevant pieces of information (SDGs)
2. Introduction (lines 25-167). As per suggestion from both reviewers, this section has been dramatically reduced from 4000 words to 2500 words, removing the least relevant information. It has been restructured
3. Lines 55-68. Addressing comments from Reviewer #2 about rationalising our use of Fe WTR and pros/cons of its use. These are also discussed in section 1.2 Reviewer #1, comment 2. We have removed the idea of a holistic approach and instead stated why we do not have a discussion about the chemical/and or biological characteristics. We have not investigated these properties and there is little literature to refer to with reference to WTR. Thus we cannot compare Al and Fe at this time.
4. Line 88. Removed "and" as per Reviewer #2
5. Line 93 Changed spelling as per Reviewer#2
6. Line 110-167 Literature review restructured and shortened as per both reviewers, to include two distinct sections. One addresses the biological/chemical implications that are most notable, the second discusses the known physical changes to soil based on WTR application.
7. Line 191-193 Reviewer #1, comment 3. As requested, information about the issues with settling is provided with reference to a thesis that details difficulties around hydrometer testing with residuals.
8. Line 235-238, text moved into this section as per point 4 from Reviewer #1
9. Line 298-305 Reviewer #1, comment 5. We have now noted our suggestions of why 30% compost behaves differently to other amendments.
10. New section added lines 346-365. This section addresses Reviewer #1, comment 6 – we are discussing all the physical properties in reference to their effect on soil health/flood holding capacity.